# Facial Expression Recognition Based on Squeeze Vision Transformer

**DOI:** 10.3390/s22103729

**Published:** 2022-05-13

**Authors:** Sangwon Kim, Jaeyeal Nam, Byoung Chul Ko

**Affiliations:** Department of Computer Engineering, Keimyung University, Daegu 42601, Korea; swkim@stu.kmu.ac.kr (S.K.); jynam@kmu.ac.kr (J.N.)

**Keywords:** facial expression recognition, vision transformer, squeeze module, visual token, landmark token

## Abstract

In recent image classification approaches, a vision transformer (ViT) has shown an excellent performance beyond that of a convolutional neural network. A ViT achieves a high classification for natural images because it properly preserves the global image features. Conversely, a ViT still has many limitations in facial expression recognition (FER), which requires the detection of subtle changes in expression, because it can lose the local features of the image. Therefore, in this paper, we propose Squeeze ViT, a method for reducing the computational complexity by reducing the number of feature dimensions while increasing the FER performance by concurrently combining global and local features. To measure the FER performance of Squeeze ViT, experiments were conducted on lab-controlled FER datasets and a wild FER dataset. Through comparative experiments with previous state-of-the-art approaches, we proved that the proposed method achieves an excellent performance on both types of datasets.

## 1. Introduction

As society continues to develop, the number of people experiencing mental and psychological anxiety in social relationships is increasing. Accordingly, there is a growing demand for improving the psychological and emotional states of individuals by recognising their level of anxiety or depression in advance. By fusing human emotions with products and services in daily life, the recognition of human emotions can be expected to create a new convergence market with various industries (e.g., medical, automobile, education, sales, and advertising). Emotion recognition can be broadly divided into bio- and image-based emotional recognition. Bio-based emotion recognition is a method used to train a computer to recognise emotions through various signals from different types of biosensors, such as an electroencephalograph [1], an electrocardiogram [2], and an electromyograph [3], and has the advantage of a high recognition rate. However, there is a limit in that a complex device must be worn, and no movements should occur during the sensing process.

Another method is facial expression recognition (FER) based on image information. FER does not require a complicated sensor to be worn; thus, it has less resistance, allowing emotions to be recognised through a camera image without the awareness of the individual. Early studies into FER extracted, analysed, and identified facial features through classical machine learning and computer vision; however, with such approaches, it was difficult to recognise facial expressions that depend on the background and image illuminance around the face. However, along with the development of convolutional neural network (CNN) algorithms, research has recently been developed to accurately recognise frequently changing dynamic facial expressions in various surrounding environments. Therefore, this study focuses on camera-image-based FER.

Studies on image-based FER have mainly focused on recognising seven human emotions: happiness, surprise, anger, sadness, fear, disgust, and neutrality. Although 22 complex emotions, e.g., ‘happily surprised’, have also been defined beyond these 7 basic emotions [4], they lack flexibility and are insufficient when elaborating on the subtleties among expressions of different moods [5]. Therefore, recognition of the seven common facial expressions or subtle facial expressions is more important for many HCI applications.

An image-based FER method mainly applies traditional machine learning and a CNN. A few studies on FER using vision transformers have recently been conducted. Traditional FER approaches typically extract features from faces and face components and train the FER classifiers, including a support vector machine (SVM) [6], AdaBoost [7], and random forest [8]. Although such an approach requires lower computing specifications and less memory than a CNN-based method, they require an expert to determine the feature extraction and classifiers, thereby lowering the performance in comparison to a CNN.

By contrast, CNN-based approaches [9,10,11,12,13], unlike traditional approaches, use deep convolutional neural networks to extract the desired features from the data and directly learn the optimal features and classifiers. Although this approach shows a high accuracy in terms of the FER in comparison to a traditional method, higher computing specifications are required to conduct the training and testing. Conversely, there is a need for a new method to reduce the computational burden in comparison to a CNN algorithm while concurrently increasing the accuracy, which has been a bottleneck with this method [14].

In the field of image recognition, ViT-based methods show a state-of-the-art (SOTA) performance in comparison to a CNN-based approach. A ViT uses not only image recognition performance but also computational efficiency and scalability; thus, it can efficiently use the memory and improve the speed. A ViT shows strength in global image classification, because it is suitable for checking the overall relationship by modelling the dependence between different parts of an image. It is, therefore, unsuitable for the classification of delicate images such as facial expressions. For this reason, relatively few FER studies using ViTs have been conducted.

In this study, we focus on the development of a ViT-based FER algorithm to enable accurate FER when applying various datasets. The contributions of this study include the following:(1)We combine existing visual tokens and landmark heatmap-based local tokens allowing the ViT to maintain the global and local features at the same time;(2)To reduce the excessive number of computations of the ViT, we propose a squeeze module to reduce the number of feature dimensions and parameters for each encoder layer;(3)To prove that the proposed Squeeze ViT is robust to FER under various environments, we measure the performance not only on lab-controlled FER datasets but also on a wild FER dataset;(4)Through various ablation studies, we prove that the FER performance increases when visual and landmark tokens are used together and demonstrate that Squeeze ViT can significantly reduce the numbers of parameters and computations in comparison to pure ViTs;(5)The remainder of this paper is structured as follows: In Section 2, we present a review of the related studies on FER based on conventional approaches, a CNN, and a ViT. Section 3 provides the details of our proposed Squeeze ViT method. Section 4 provides a comprehensive evaluation of the proposed method based on various experiments. Finally, some concluding remarks are given in Section 5.

## 2. Related Studies

The FER-related studies considered herein can be divided into existing machine learning methods, a CNN-based method achieving an SOTA performance, and a ViT-based method, which is a recently emerged approach in the field of image classification. In this section, we introduce detailed algorithms for each category.

### 2.1. FER Based on Conventional Approaches

A common approach to traditional machine-learning-based FER systems is detecting facial regions and extracting geometric features, appearance features, or hybrid geometric and appearance features from the target face [14]. Most methods using geometric features extract facial landmarks and learn classification algorithms based on changes in the distance and angle between landmarks reflecting changes in the facial expressions. Jeong et al.’s [8] method detects facial landmarks in an input image and extracts geometric features by considering the spatial locations between landmarks. The extracted feature vectors are then applied to their proposed hierarchical weighted random forest classifier to classify the facial expressions. In addition, Ghimire and Lee’s [7] method similarly extracts two types of geometric features based on the positions and angles of 52 facial landmark points and uses a two-way classification method applying multi-class AdaBoost and an SVM based on the boosted feature vectors. As an FER method using appearance, Greche et al. [15] also presented an FER based on three steps, i.e., data preparation, feature extraction using a histogram of gradient, and template matching for classification using a normalised cross-correlation. In addition, Lee et al. [16] proposed a new sparse expression-based FER method to reduce the within-class variation while emphasising facial expressions in query face images. To this end, a new method was presented for generating intraclass transformed images of each representation using training representation images. Ghimire et al. [6] also proposed a method for recognising FER in a single image frame using a hybrid combination of the appearance and geometric features and applying an SVM classifier. Hybrid FER methods compensate for the weaknesses of individual approaches and provide better results in certain cases.

### 2.2. FER Based on CNN

Unlike traditional machine-learning-based FER, as the main advantage of CNN-based FER, it enables ‘end-to-end’ learning directly from the input images, completely eliminating or significantly reducing the dependence on physically based models or other pre-processing techniques [14]. Zhao et al. [9] proposed a feature selection network that automatically extracts and filters facial features by embedding a feature selection mechanism inside AlexNet [17]. In the feature selection mechanism, the proposed network was designed to effectively remove extraneous features according to the learned feature map and enhance the FER performance on deformed facial expressions. In addition, Fan et al. [10] proposed a deep-supervised attention network that captures more identifiable information from deep layers to shallow layers by utilising multi-scale features in the face images. In addition, this method combines the complementary features of multiple convolutional layers in a deep supervised manner, and a method is proposed for using ensembles of the intermediate prediction scores. Numerous attention-mechanism-based FER methods that learn a feature map to focus on a part of the face and use it for expression recognition have been studied. Representatively, Xu et al. [11] proposed solving the bias using a facial attention attribute as the input; then, they conducted experiments on the bias and fairness for FER. Wang et al. [12] proposed a self-curing network for FER that takes full advantage of an attention mechanism to further suppress the uncertainty contained in the dataset and give a weight each training face sample. In addition, Minaee et al. [13] proposed a deep learning approach based on attentional convolutional networks that can focus on important parts of the face and achieve significant improvements over previous models on multiple datasets. To date, in FER research, CNN-based methods have mainly been studied, although they have tended to reach their performance limit.

### 2.3. FER Based on ViT

Inspired by the successful results of applying a transformer in natural language processing (NLP), the ViT [18] was proposed in the field of computer vision in 2021 and has shown excellent performance in image classification apart from CNN-based feature expression techniques. Although the ViT has begun to be applied to various vision problems such as object detection [19], object tracking [20], and image synthesis [21], as well as image classification [18], there are few cases in which it has been applied to FER.

To demonstrate the effect on FER, Xue et al. [22] proposed an algorithm called ViT-FER that adaptively characterises the relationship between different face parts by applying a ViT. Aouayeb et al. [23] proposed training a ViT using a squeeze and excitation block for FER tasks. This method also attempts to improve FER performance by providing an internal expression analysis of the ViT for facial expressions. Wang et al. [24] proposed a progressive multi-scale ViT (PMVT) for capturing complex relationships between different facial action units (AUs) for different expressions in a data-driven manner. The PMVT is based on a multiscale self-attention mechanism that can flexibly focus on a series of image patches and encode important signals for the AUs. In addition, Ma et al. [25] proposed a ViT with a feature fusion that consists of two main steps for FER. By fusing global–local attention with multiple functions, the identified information is captured, and the fused feature map is flattened and projected into a visual word sequence. The relationship between visual words is modelled as a global self-attention and applied to FER.

Because facial expressions need to detect subtle changes in muscles, if the pure structure of the ViT, which does not reflect the local features, is used as is, the performance may be lower than that of CNN-based FER. Therefore, a novel ViT structure that can reflect subtle changes and feature attention in facial expressions should be proposed.

## 3. Proposed Method

In this section, we propose Squeeze ViT for generating new output features reflecting both local and global features; then, we describe an FER algorithm that can recognise more subtle changes in facial expression based on these features.

### 3.1. Overview of Squeeze ViT

In this study, we designed the FER model by simultaneously utilising the input face and facial landmarks. As shown in Figure 1a, the input image is fed to ResNet-18 f(⋅) to generate a local-level feature map z1∈ℝc1×h1×w1 and a global-level feature map z2∈ℝc2×h2×w2. Here, c*, h*, and w* represent the channel, width, and height of the *-layer feature map. Specifically, z1 and z2 refer to the third and fourth module outputs of f(⋅), respectively. At the same time, we extract landmarks using FaceAlignment [26], and each keypoint of the facial landmarks K is used for creating NK heatmaps for each landmark keypoint M∈ℝNK×h1×w1. To this end, we generate NK (same number) landmark tokens TK∈ℝc1×NK from the computations of z1 and heatmaps M. The global-level feature map z2 is flattened into NV visual tokens TV∈ℝc2×NV through a tokenizer. These tokens are concatenated and then used to predict the final label through a series of transformer layers. We apply an efficient model design to alleviate the computational complexity, which is a disadvantage of existing transformers. As shown in Figure 1c, the proposed model uses the squeeze modules of Figure 1b across multiple stages to gradually reduce the number of feature dimensions while maintaining robust features, mitigating the trade-off between high performance and efficiency. The architecture of the squeeze module is similar to CBAM [27], designed for channel-wise attention in the existing ResNet model, but the proposed squeeze model is simply designed for reducing the feature dimension, and the attention of tokens is obtained using a ViT.

### 3.2. Review of ViT

The ViT [18] is a meaningful approach to image processing without significantly changing the overall architecture of the transformer [28]. The ViT has shown a performance exceeding that of CNN-based models by using a transformer that applies a sequence of image patches as an input without relying on a CNN.

To create the input data, the image is divided into patches, each of which is arranged in order from the top left to the bottom right to create a data sequence. Each patch is flattened and converted into a vector. Each vector is then embedded through a linear operation, and a class token that predicts the class is added. By adding positional embedding to the input vector to which the class token is added, the input of the ViT is complete. After the embedded patch matrix z is normalised through LayerNormalisation (LN), it is applied to the transformer encoder composed of multi-head attention (MSA) and then fed to the multi-layer perceptron (MLP) to make the output vector have the same size as the input vector through L iterations. The output y of the transformer encoder also consists of a class token and a vector. Here, we can finally classify the image by applying it to the MLP using only the class token:(1){Q=zWQ∈ℝ|z|×dK=zWK∈ℝ|z|×dV=zWV∈ℝ|z|×d
(2)MSA(z)=softmax(QKTd)V
(3)MLP(z)=GeLU(∑inwiz+b)
(4)zl′=MSA(LN(zl−1))+zl−1,l=1…L
(5)zl=MLP(LN(zl′))+zl′,l=1…L
(6)y=LN(zL0)

### 3.3. Refined Representation of Tokens

Although the ViT has shown excellent performance in terms of global modelling by feeding images as sequential patches, it does not reflect the strong local structure of the images, because it treats all image patches (tokens) equally and ignores the locality. In FER, global-level features support a high-level image abstraction, which is useful for learning the relationships between global features within face regions. However, there is a limit to using only the global facial features, because local features that are emphasised when making a facial expression have a greater effect on the FER performance. Therefore, to achieve a high performance in FER tasks, we combine global- and local-level features as the ViT input.

First, global-level feature z2∈ℝc2×h2×w2, which is sufficiently abstracted by multiple layers of the ResNet-18 backbone, takes the form of *N* visual tokens, TV∈ℝc2×NV, through a simple tokenizer. Using the squeeze module, the visual token TV is projected into a latent feature space and output as T^V∈ℝd×NV, where d is the number of dimensions of the latent feature space, and NV=h2×w2 is the number of visual tokens.

We designed the squeeze module to project different feature spaces into the same latent feature spaces, as shown in Figure 1c. The squeeze module outputs the normalised attention by successively applying global average pooling (GAP), a fully connected (FC) layer, ReLu activation function, a second FC layer, and, finally, the sigmoid function to the input tokens. Attention is again concatenated with the original input tokens and transformed into a latent feature space through the FC layer.

Unlike a global-level feature, the local-level feature z1∈ℝc1×h1×w1 is tokenized into TK∈ℝc1×NK using a heatmap. The process of extracting local-level features from the heat map is as follows: First, we acquire NK= 68 keypoint coordinates from facial landmarks using FaceAlignment [26]. The facial landmark set K∈ℕ2×NK is constructed by the x- and y-coordinates of keypoints. Second, a total of 68 heatmaps M∈ℝ68×h1×w1 with spatial dimensions of z1 are generated using the keypoint coordinates of the landmark. Third, the value of the heatmap is normalised from 0 to 1. Using these coordinates, the tokenization of the local-level features is as follows:(7)TK(k)=∑h∑wM(k,h,w)z1(:,h,w),k∈{1,…,NK}
Different from a visual token, the landmark token TK also needs to be projected into the same latent feature space for transformer training. A squeeze module for landmark tokens converts the tokenized local-level features into a latent feature space using weights differing from the visual tokens; then, it outputs them as T^K∈ℝd×NK.

From visual token T^V and landmark token T^K, we further define the final transformer token T by applying the [CLS]∈ℝd×1 token and absolute position encoding (pos) for the final label prediction, as follows:(8)T=[T^V,pos([T^K,CLS])]∈ℝd×(NV+NK+1)

### 3.4. Squeeze ViT

An overview of the Squeeze ViT architecture is shown in Figure 1c. Each layer consists of a transformer encoder and a squeeze module. The proposed Squeeze ViT has the following characteristics compared with the existing ViT-based models:(1)ViT can learn the global expressions more easily than a CNN by using self-attention for all tokens; however, this characteristic is rather disadvantageous for tasks in which local representation is important, such as FER. To alleviate this limitation, we allow the transformer encoder to interact between mixed (global + local) representation tokens;(2)Existing ViT [18]—based transformers [29,30] encode tokens with the same transformer layer stack under the same network settings. Therefore, the input and output tokens share characteristics with the same number of tokens. Although these methods are simple and effective, to achieve good performance, they are computationally expensive. The squeeze module can also reduce the parameters and operations of subsequent transformer layers by progressively reducing the number of feature dimensions of the token. In this process, although a feature loss may occur as the feature dimension of the token is reduced, such a loss can be overcome using feature-wise attention in the squeeze module;(3)With the ViT, the output of the last transformer encoder is used for classification, whereas with Squeeze ViT, the output of the last squeeze module is used for classification. The class token of the final output is applied to the MLP to finally classify the facial expressions.

The detailed procedures used for training the proposed Squeeze ViT are described in Algorithm 1.
**Algorithm 1:** Training of Pytorch-like Squeeze ViT Structure# f: backbone returning intermediate layer outputs (third and fourth)# s: squeeze module# pos: absolute position encoding# layers: list of tuples (e, transformer encoder; s, squeeze module)# h: prediction headcls = Parameter()for (x, y, M) in loader:    z1, z2 = f(x) # z1, local feature map; z2, global feature map    t1, t2 = K(z1, M), G(z2) # t1, landmark tokens; t2, visual tokens    t1, t2 = s(pos(t1)), s(t2) # projections, c1-by-d and c2-by-d, respectively    t = cat(t1, t2, cls)    for (e, s) in layers: # processing of transformer encoder (e), squeeze module (s)        t = e(t) # embedding tokens        t = s(t) # projection, d-by-d′    p = h(t.cls) # projection head, only for the CLS token    L = LabelSmoothingCrossEntropy(p, y) # loss function    L.backward() # back-propagate    update(f, layers, h) # AdamW updatedef K(z, M): # generation of landmark tokens    z = z.expand() # reshape, (c1,h1,w1)-by-(1,c1,h1,w1)    M = M.expand() # reshape, (68,h1,w1)-by-(1,68,h1,w1)    t = z^T^M # dot prod, (1,c1,h1,w1)(1,68,h1,w1)-by-(c1,68,h1,w1)    t = sum(t) # reduce sum, (c1,68,h1,w1)-by-(c1,68)    return tdef G(z): # generation of visual tokens    t = z.flatten() # flatten, (c2,h2,w2)-by-(c2,h2×w2) return t

## 4. Experimental Results

In this section, we introduce the datasets mainly used in FER experiments and prove that the proposed method is superior to other methods for both wild and lab-controlled datasets through comparative experiments conducted using the latest FER-related approaches.

### 4.1. Dataset

During the experiment, to prove that the proposed method shows a highly robust performance on various datasets, we used the CK+ and MMI datasets as lab-controlled datasets and RAF-DB as a wild dataset.

**CK+ [31]**: CK+ is the most widely used database in FER, consisting of 593 video sequences and expression labels for 123 individuals aged from 18 to 30. Seven emotions labels are applied: anger, contempt, disgust, fear, happiness, sadness, and surprise. Because CK+ data do not provide separate training and test data, in this study, a subject-independent 10-fold cross-validation was applied, and the accuracy was measured. The pixel resolution of the image was 640 × 480 and 640 × 490 in the grey scale.

**MMI [32]****:** MMI contains 2900 videos and high-resolution still images of 75 subjects, including frontal and lateral faces. The image sequence contains six basic facial expressions, with the states of the facial expressions starting with neutral, followed by apex, and ending with neutral again. Because this database does not provide the location of the peak frames, we used three randomly collected apex frames with the six basic emotion labels provided, similar to the experiment conducted by Jeong et al. [8] The experiment used the same subject-independent 10-fold cross-validation used with CK+. Each face was 720 pixels × 576 pixels in size.

**RAF-DB** [33]: The representative wild FER dataset, the Real-world Affective Face Database (RAF-DB), contains a total of 29,672 face images labelled by 40 trained taggers. The face images are labelled with 7 basic emotions and 11 complex emotions. This database contains the following variations for diversity: faces transformed through post-processing operations using different ages, genders, head poses, lighting conditions, occlusions by glasses and hair, special effects, and filtering. During the experiment, 12,271 out of 15,339 images in the basic emotion set were used as training data, and the remaining 3068 images were used as test data.

### 4.2. Implementation Details

During our experiments, facial landmarks were extracted in advance using FaceAlignment [26] on a face image adjusted to a pixel resolution of 224×224. The facial features were extracted from the last and prior to the last convolution steps of ResNet-18. The learning rate of our method was initialised to 0.005, and we used a cosine scheduler. The AdamW optimiser was applied and trained for 200 epochs in CK+ and MMI and for 300 epochs in RAF-DB, with a batch size of 256. Label smoothing cross-entropy loss was used as the overall training loss function. We implemented the model using the Pytorch framework and conducted all experiments on four V100 GPUs.

### 4.3. Ablation Studies

An important question is whether combining each token is useful. In existing ViT-based models, tokens composed of only visual representations have been employed. Other tasks such as classification, object detection, or segmentation can achieve a dominant performance based on visual representational learning. However, in the case of FER, both visual features and localised features, e.g., landmark points containing an emotional representation, may be important for recognising detailed emotions. To compare the difference in performance between the use of only visual tokens and landmark tokens together, we applied two inputs into the same Squeeze ViT model. The experiment was conducted on the wild RAF-DB dataset with large changes in the facial characteristics. The experimental results listed in Table 1 show that adding landmark tokens improved the performance by 0.9% compared with using only visual tokens. This clearly shows that the performance bottleneck of the proposed Squeeze ViT model can be solved using landmark tokens rather than single visual tokens.

In an additional experiment, we tested how much Squeeze ViT can reduce the numbers of parameters and computations in comparison to the pure ViT. As the experimental results in Table 2 show, Squeeze ViT reduced the number of parameters by approximately 7.26-fold and the number of computations by approximately 17.95-fold. In particular, because the squeeze module reduces the number of feature dimensions for each encoder, the rate of reduction in the number of computations was larger than that of the number of parameters. As the experimental results indicate, the proposed Squeeze ViT approach is much more efficient in terms of memory, operation speed, and FER than the pure ViT.

### 4.4. Performance Comparison with Previous State-of-the-Art Methods

Based on various ablation studies, we proved that Squeeze ViT can significantly reduce the numbers of parameters and computations compared with the pure ViT. In this section, we also prove that Squeeze ViT can increase the FER performance when visual and landmark tokens are used together on the lab-controlled CK+ and MMI datasets, as well as on the wild RAF-DB dataset. In the comparative experiment, the FER accuracy was measured for basic emotion classes (seven emotions for CK+ and RAF-DB and six emotions for MMI).

The SOTA methods used in the comparative experiment are as follows: (1) a hierarchical weighted random forest (WRF)-based FER classifier [8], (2) inconsistent pseudo annotations to latent truth (IPA2LT) [34], (3) facial motion prior networks (FMPN) [35], (4) auxiliary label space graphs (ALSG) [36], (5) feature decomposition and reconstruction learning (FDRL) [37], (6) local non-local joint network (LNLAttenNet) [38], (7) vision transformer jointly with squeeze and excitation (ViT-SE) [23], (8) distribution mining and pairwise uncertainty estimation (DMUE) [39], (9) relative uncertainty learning (RUE) [40], and (10) the proposed Squeeze ViT algorithm.

As shown in the experimental results listed in Table 3, the proposed Squeeze ViT method showed the best performance on the CK+ and MMI datasets. In addition, FDRL [37] showed a similar performance as the proposed method on CK+ but a 4.66% lower performance on MMI. However, for the wild dataset RAF-DB, FDRL [37] showed a 0.57% higher performance than the proposed method. It was determined that the recognition performance of FDRL [37] was improved because the additional MS-Celeb-1M face recognition database [41] was used for the pre-training of ResNet-18, the backbone network of FDRL. WRF, a traditional machine learning technique, achieved a relatively poor performance compared with the CNN-based FER methods. In the case of DMUE and RUL applied only for to the wild FER, a deep network based on uncertainty learning was used but demonstrated results lower (DMUE, 0.14%) or slightly higher (RUL, 0.08%) than those of the proposed method. ViT-SE [23] based on the ViT structure showed a lower performance than the proposed method on both the CK+ and RAF-DB datasets.

Based on the experimental results, we can see that the proposed Squeeze ViT approach exhibits a uniformly excellent performance on not only well-controlled indoor facial expressions but also on wild facial expressions.

### 4.5. Confusion Matrix Comparison

Considering that the facial expressions in the wild RAF-DB dataset have an imbalanced distribution, the accuracy has certain limitations in terms of the analysis for imbalanced data. Therefore, we further examined the comparative performance based on confusion matrices. In Figure 2, the diagonal entries represent the recognition accuracy for each expression. We present the following results:i.**CK+**: The proposed model showed a degraded performance on a *surprise* (*SU*) expression, which is difficult to distinguish from *disgust* (*DI*) and *neutral* (*NE*). Regardless, it showed a perfect probe performance on *disgust* (*DI*), *happy* (*HA*), and *sad* (*SA*) expressions;ii.**MMI**: The overall performance is good except for *angry* (*AN*), *fear* (*FE*), and *sadness* (*SA*). In the case of *fear* (*FE*), a lower accuracy was demonstrated because *fear* (*FE*), *angry* (*AN*), and *surprise* (*SU*) share relatively similar muscle movements as the other expressions;iii.**RAF-DB**: Similar to the other datasets, the proposed model showed a relatively good performance for *happy* (*HA*) and *sad* (*SA*). By contrast, *disgust* (*DI*) seemed to be easily misclassified as *sadness* (*SA*) or *anger* (*AN*). Finally, *neutral* (*NE*) was also confused with *sadness* (*SA*).

## 5. Conclusions

In this study, we propose a new type of Squeeze ViT method using a vision transformer as the backbone to improve FER performance. To improve the problem in which ViT represents the global features well but does not support the local features, we propose a feature representation method that can consider global- and landmark-oriented local features at the same time. In addition, to reduce the number of large parameters and the computational complexity, which are problems with pure ViT, a squeeze module is proposed, and the number of feature dimensions is adjusted for each encoder layer. Based on the experimental results, we confirmed that the performance was remarkably improved when the visual and landmark tokens were used at the same time. In addition, the proposed method showed an excellent performance on both the wild dataset and the lab-controlled datasets in comparison with the other SOTA FER methods. The proposed method showed a slightly lower performance than the FDRL [37] method for RAF-DB, which is due to RAF-DB being a wild dataset; thus, the landmarks could not be accurately extracted owing to extreme lighting conditions, occlusions, and severe changes in pose. Therefore, in a future study, we plan to expand this research work to include the extraction of features by estimating landmarks or facial patches even in occluded face regions.

## Figures and Tables

**Figure 1 sensors-22-03729-f001:**
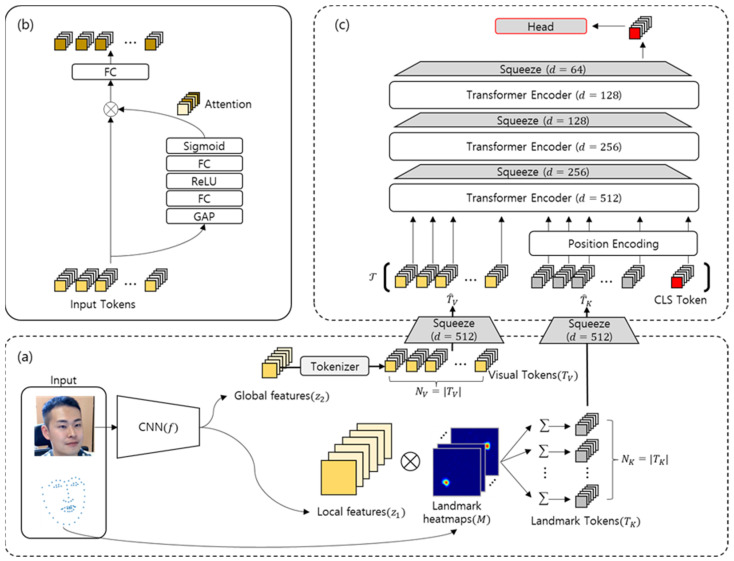
Illustration of the proposed Squeeze ViT. (**a**) An input image is fed to the CNN to yield the global and local features $z_{1}$ and $z_{2}$. Each feature is transformed into visual tokens $T_{V}$ and landmark tokens $T_{K}$, respectively. (**b**) The squeeze module adjusts the feature dimensions while maintaining robust discriminative elements. (**c**) Concatenated tokens T are fed to the Squeeze ViT having multiple stacks of encoders and squeeze modules to reduce the computational complexity.

**Figure 2 sensors-22-03729-f002:**
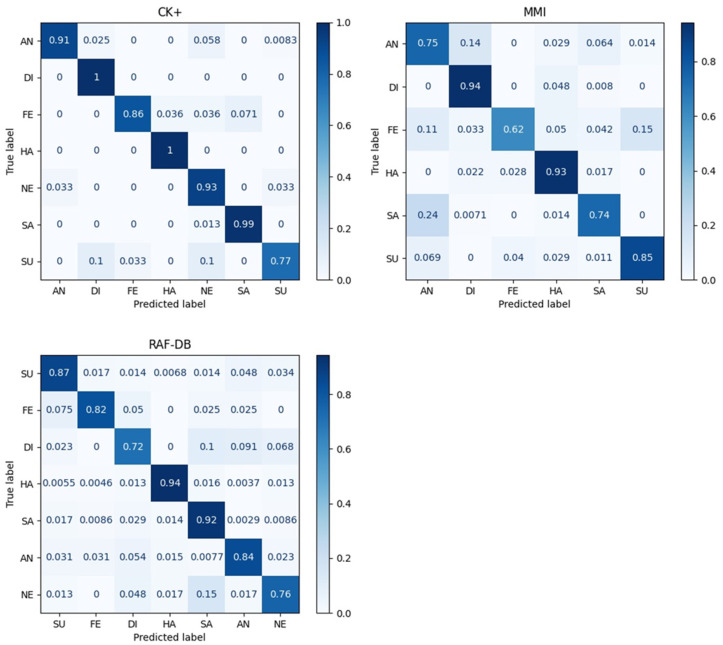
Confusion matrices obtained by the proposed Squeeze ViT approach using three datasets. **CK+** and **MMI**: Results accumulated across all 10-fold cross-validations. **RAF-DB**: Results from RAF-DB validation set.

**Table 1 sensors-22-03729-t001:** Ablation study of tokens for each configuration on the RAF-DB dataset.

Visual Token	Landmark Token	Accuracy (%)
✔	✘	88.0
✔	✔	88.9

**Table 2 sensors-22-03729-t002:** Comparison results on the numbers of parameters and computations between pure ViT and the proposed Squeeze ViT method on the CK+ dataset.

Methods	Params (M)	FLOPs (G)	Accuracy (%)
ViT [18]	86.86	33.03	98.75
Squeeze ViT	11.96	1.84	99.54

**Table 3 sensors-22-03729-t003:** Performance comparison with state-of-the-art methods applied to the CK+, MMI, and RAF-DB datasets. Bold marks the best accuracy.

Methods	Accuracy (%)		
	CK+	MMI	RAF-DB
WRF [8]	92.6	76.7	-
IPA2LT [34]	91.67	65.61	86.77
FMPN [35]	98.06	82.74	-
ALSG [36]	93.08	70.49	85.53
FDRL [37]	**99.54**	85.23	**89.47**
LNLAttenNet [38]	98.18	68.75	86.15
ViT-SE * [23]	99.49	-	87.22
DMUE [39]	-	-	88.76
RUL [40]	-	-	88.98
Squeeze ViT	**99.54**	**89.89**	88.90

* ViT-SE refers to the test results from [23] under the same conditions as those used in our study.

## Data Availability

Not applicable.

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
