# Peer review of "Facial Expression Recognition Based on Squeeze Vision Transformer"

_sensors, 2022, doi:10.3390/s22103729_

Round 1
Reviewer 1 Report
The paper is quite interesting. A lot of details about the methodology are presented.
Author Response
Thanks for your valuable comments.
Reviewer 2 Report
A new type of Squeeze ViT method is proposed to improve the FER performance. ViT has shown an excellent performance in image classification. However, ViT still has many limitations in FER, which requires the detection of subtle changes in expression because it can lose the local features of the image. Therefore, the paper proposed Squeeze ViT, a method for reducing the computational complexity by reducing the number of feature dimensions while increasing the FER performance by concurrently combining global and local features. But to a better paper, there are some problems should be revised.
- In line180. Why can CNN generate two types of features? If it can generate global features, why do you need ViT? Please reference papers:
- a) Occlusion-Adaptive Deep Network for Robust Facial Expression Recognition
- b) A convolution-transformer dual branch network for head-pose and occlusion facial expression recognition
- What is the difference between the squeeze module you designed and the article “Cbam: Convolutional block attention module”?
- Explain in detail the process of obtaining heatmaps from landmarks.
- As far as I know, both CK+ and MMI do not divide training set and test set. Therefore, many studies will conduct experiment with 10-fold-subject-independent strategy, what is your method? Please reference papers:
- a) Facial expression recognition using temporal POEM features
- b) Facial Expression Recognition via Deep Action Units Graph Network Based on Psychological Mechanism
- In Table 2, only use pure ViT can achieve the accuracy of 98.75%? On CK+ or MMI? Can you elaborate on this process? I'm guessing you are using a randomly divided dataset!
Reviewer 3 Report
This paper proposes a face expression recognition based on a deep learning approach. The evaluation results show that the proposed scheme performs better than other previously proposed schemes. However, several issues must be attended before to take a decision about a potential acceptation of the paper
- It would be convenient to include references of ViT in some indexed journals instead or beside the published in arXiv sources.
- Please explain how the tokens are obtained and how the tokens are used.
- It is necessary to mention how the functions in equations (1)-(3). For example MSA(LN(z)), MLP(LN(z)). Please define the functions used.
- Define the variables used in eq. (4).
- It is necessary to include a detained explanation of the proposed ViT structure shown in Fig. 1. It is necessary the provide more details of these system. The operation performed by the block diagrams of Fig. 1(a)-1(c).
- The size of Fig. 2, must be increased because the fonts are to small and it is not possible to read them.
Round 2
Reviewer 2 Report
The design of the experiment is unreasonable. 10-fold cross-validation used in the paper is random split. However, authors compared with the results of subject-independent method in Table 3, which is unfair. The most important is that subject-independent N-fold cross-validation is more in line with the experimental design.
Reviewer 3 Report
The authors attended the reviewer observations then I consider taht it can be accepted in its actual form
Author Response
Thanks for your valuable comments.